# Pectinases Secretion by *Saccharomyces cerevisiae*: Optimization in Solid-State Fermentation and Identification by a Shotgun Proteomics Approach

**DOI:** 10.3390/molecules27154981

**Published:** 2022-08-05

**Authors:** Matheus Mikio Takeyama, Márcia Corrêa de Carvalho, Helena Sacco Carvalho, Cristiane Rodrigues Silva, Ana Paula Trovatti Uetanabaro, Andrea Miura da Costa, Joseph A. Medeiros Evaristo, Fábio César Sousa Nogueira, Ana Elizabeth Cavalcante Fai, Maria Gabriela Bello Koblitz

**Affiliations:** 1Food and Nutrition Graduate Program (PPGAN), Federal University of the State of Rio de Janeiro (UNIRIO), Rio de Janeiro 22290-240, RJ, Brazil; 2Nutrition School, Federal University of the State of Rio de Janeiro (UNIRIO), Rio de Janeiro 22290-240, RJ, Brazil; 3Department of Biological Sciences, Santa Cruz State University (UESC), Ilhéus 45662-900, BA, Brazil; 4Laboratory of Proteomics/LADETEC, Institute of Chemistry, Federal University of Rio de Janeiro, Rio de Janeiro 21941-598, RJ, Brazil; 5Proteomics Unit, Institute of Chemistry, Federal University of Rio de Janeiro, Rio de Janeiro 21941-909, RJ, Brazil; 6Department of Basic and Experimental Nutrition, Rio de Janeiro State University (UERJ), Rio de Janeiro 20550-013, RJ, Brazil

**Keywords:** response surface methodology, solid-state fermentation, carbohydrate active enzymes, pectinolytic enzymes, nano LC-MS/MS analysis

## Abstract

A sequential design strategy was applied to optimize the secretion of pectinases by a *Saccharomyces cerevisiae* strain, from Brazilian sugarcane liquor vat, on passion fruit residue flour (PFRF), through solid-state fermentation (SSF). A factorial design was performed to determine the influence variables and two rotational central composite designs were executed. The validated experimental result was of 7.1 U mL^−1^ using 50% PFRF (*w*/*w*), pH 5, 30 °C for 24 h, under static SSF. Polygalacturonase, pectin methyl esterase, pectin–lyase and pectate–lyase activities were 3.5; 0.08; 3.1 and 0.8 U mL^−1^, respectively. Shotgun proteomics analysis of the crude extract enabled the identification of two pectin–lyases, one pectate–lyase and a glucosidase. The crude enzymatic extract maintained at least 80% of its original activity at pH values and temperatures ranging from 2 to 8 and 30 to 80 °C, respectively, over 60 min incubation. Results revealed that PFRF might be a cost-effective and eco-friendly substrate to produce pectinases. Statistical optimization led to fermentation conditions wherein pectin active proteins predominated. To the extent of our knowledge, this is the first study reporting the synthesis of pectate lyase by *S. cerevisiae*.

## 1. Introduction

Pectinases are important enzymes to the industry and stand out due to their increasing demand that stems from their wide application in the manufacture of agro-industrial products [1]. Pectinolytic enzymes are applied in various processes in the food industry, such as fruit juice extraction and clarification; grape pretreatment for winemaking; maceration and liquefaction of plant tissues, extraction of oil from oleaginous fruits; acceleration of tea, coffee and cocoa fermentation; reduction of bitterness in citrus peels, among many others [2,3,4].

Although pectinases are produced by plants and microorganisms, there is a greater interest in pectinases secreted by the latter. Some advantages of the microbial sources are the possibility of secretion induction by cultivation media, oriented biosynthesis with specific catalytic sites and large-scale production associated with yield optimization and productivity, regardless of seasonal factors or high-cost substrate [3,5,6]. Yeasts have drawn attention in the past years as a consequence of the diversity of pectinases that can be produced by this fungal group and their biochemical plasticity, simplifying cultivation and handling [1,7]. Among the yeast’s genera, *Saccharomyces, Kluyveromyces, Cryptococcus, Rhodotorula, Aureobasidium, Candida* and *Metschnikowia* are noteworthy for their pectinase secretion capability. The production of pectinolytic enzymes by yeasts, mainly by *S. cerevisiae*, has been targeted in the later years, aiming to establish optimized, economic and sustainable fermentation processes [1,8], as this species has been thoroughly studied and thrives even through adverse industrial conditions [9].

SSF may be an interesting alternative to overcome the challenge of cost-effectiveness in enzyme production. When compared to submerged fermentation, SSF results in higher enzyme concentration and a possibility to use low-cost growth media—such as agro-industrial residues—reducing the environmental impact and adding value to the residue [1,10]. Among the residues used as alternative substrates for pectinases production, wheat bran, banana, lemon, orange, and passion fruit peels; apple core; seedless sunflower heads; sugarcane bagasse and coffee pulp were already reported in the literature [11]. According to the FAO Food Outlook from July 2018 [12], a little under 1500 thousand tons of passion fruit are produced per year worldwide. Brazil is the largest producer, with around 950,000 tons/year. Over 40% of the production is destined to the juice industry and around 50% of the fruits total mass is discarded as peel, which may be dried, ground and marketed as a flour PFRF [13,14]. Passion fruit peel shows a pectin content of around 7 g 100 g^−1^, considerably higher than apple, persimmon, beet pulp, orange, strawberry, tomato, and grape, of which agro-industrial residues contain between 0.2 and 4 g 100 g^−1^ of pectin [15].

Considering the above, this study aimed to optimize the secretion of the pectinolytic enzyme-complex by a previously selected *S. cerevisiae* strain, on PFRF, using sequential multivariate experimental designs, as well as to characterize the enzymatic extract obtained in the validated optimized culture conditions. As the presence of several pectinolytic enzymes in the crude extract was evidenced, a shotgun proteomics approach was applied, for the first time to the extent of our knowledge, to identify the set of proteins, specifically different enzymes.

## 2. Material and Methods

### 2.1. Microorganism

Yeasts from the Microorganisms Collection of the State of Bahia, isolated from Brazilian sugarcane liquor (cachaça) vats and identified as *S. cerevisiae* by analyzing their respective DNA sequences representative of the D1/D2 region of the 26S gene from ribosomal DNA according to [16], were kindly donated and previously selected as potential pectinases producers [17]. Among the strains tested, the best pectinases producer was selected for these optimization assays, and the culture was kept in microtubes with 80% of saline solution (0.9% NaCl) and 20% of sterile glycerol and stored in a freezer at −80 °C (Indrel, IULT 335 D, Londrina, PR, Brazil).

### 2.2. Sequential Strategy of Experimental Designs for the Production of Pectinases by SSF

#### 2.2.1. Culture Medium and Inoculum

The assays consisted of 250 mL conical Erlenmeyer flasks containing 20 g of culture medium, whose composition varied according to the experimental design, with the addition of water (pH 7) or buffer (pH 5, 7, and 9) to different amounts of peptone and PFRF, to reach the concentrations (% *w*/*w* of medium) shown in Table 1. Depending on the proportions of PFRF and water/buffer used, the culture medium could turn out liquid or solid, generating a submerged fermentation process or a SSF process. The flasks containing the medium were capped with cotton and gauze and sterilized under pressure for 15 min at 121 °C. A standard yeast cell suspension was prepared using turbidity as determined by the absorbance at 600 nm (Shimadzu spectrophotometer, UV-VIS 2700, Japan) from the culture cultivated overnight in Sabouraud agar medium (Kasvi, São José dos Pinhais, PR, Brazil) and progressively adjusted with saline solution to 0.7 absorbance units, which corresponded to 10^8^ cells/mL [18]. To each assay, 2 mL (0.1 mL/g) of standard cellular suspension was inoculated, and the experiments were kept in a rotatory incubator (Nova Técnica, Incubadora Shaker Refrigerada NT 715, Piracicaba, SP, Brazil), at controlled temperature and under agitation or statically, according to the experimental design (Table 1).

#### 2.2.2. Experimental Designs

A sequential strategy of experimental designs was applied to optimize the SSF process, aiming for increased pectinases secretion [19]. Initially, a fractional factorial design (2^n−1^) was performed, in which five independent variables (pH, agitation, temperature and concentration of peptone and PFRF) were evaluated in three levels. The ranges studied were selected based on the literature and are shown in Table 1.

After determining the significant variables, (*p* < 0.05), two central composite rotational designs (CCRD) were performed in sequence, with three (peptone and PFRF concentrations and temperature) and two independent variables (PFRF concentration and temperature), respectively (Table 1). The dependent variable (response) analyzed in all designs was the total pectinolytic activity (U mL^−1^), determined according to the specified in ‘Enzyme Assays’ below. The data analysis was performed using the software STATISTICA (version 7.0) [20].

After the assays proposed by the sequential strategy of experimental designs, the process was validated in order to verify the adequacy of the model. Thus, independent experiments were performed in triplicates under optimized conditions. The values mathematically predicted by the equation generated in CCRD-2 were compared to those obtained with experiments in the process validation using the Student’s *t*-test (*p* < 0.05).

#### 2.2.3. Crude Enzymatic Extract

The recovery of the crude enzymatic extract for the fractional factorial design was performed by adding 0.5 mL of distilled water per gram of wet basis fermentation medium, and for CCRD-1 and CCRD-2, 1 mL of acetate buffer solution (0.2 mol L^−1^, pH 5) per gram of culture medium. The culture medium was centrifuged (Heraeus Megafuge, 16 R, Canada) at 4 °C, 3500× *g*, for 15 min, for precipitation of the cellular mass and remaining PFRF, and the supernatant (crude enzymatic extract) was aliquoted and frozen (−20 °C).

#### 2.2.4. Enzyme Assays

Total pectinolytic activity was evaluated according to the method described by [21]. One unit of pectinase activity (U mL^−1^) was defined as the amount (µmol) of mono galacturonic acid released by 1 mL of enzymatic crude extract per min. The values mathematically predicted by the Equation (1) [22]:(1)$$\mathrm{Pectinase}\;\mathrm{activity}\;\left(U\;{\mathrm{mL}}^{-1}\right)={\frac{\left[\mathrm{RS}\right]\times V_{r}}{V_{\mathrm{enzymatic}\;\mathrm{extract}}\times t\times M.W.}}^{}$$
where,

[RS] = concentration of reducing sugar (mg/mL);

V_r_ = reaction volume (mL);

V_enzymatic extract_ = volume of enzymatic extract (mL);

t = reaction time (min);

MW = molecular weight of the reducing sugar (mg µmol^−1^).

Polygalacturonase (PG) activity was determined according to the method adapted from [23]. One unit of polygalacturonase was defined as the amount (µmol) of mono galacturonic acid released by 1 mL of enzymatic crude extract per min. The PG activity was calculated according to Equation (1).

Pectin methyl esterase (PME) activity was determined according to [24]. One unit of pectin methyl esterase was defined as the amount (µmol) of mono galacturonic acid released by 1 mL of enzymatic crude extract per min. The PME activity was calculated according to Equation (1).

Pectin lyase (PL) activity and Pectate lyase (PTL) activity were determined according to the adapted procedure of [1]. One unit of activity was defined as the amount of enzyme needed to increase the medium absorbance in 1 × 10^−3^ absorbance unit. per min, per mL 

#### 2.2.5. Effect of pH and Temperature on the Crude Extract Activity and Stability

For optimum pH evaluation, the pH of the reaction media, as described above, ranging from 2 to 8 (pH 2—potassium hydrochloride buffer, pH 3—glycine, pH 4 to 5—sodium acetate and pH 6–8—sodium phosphate, all in a concentration of 0.2 mol L^−1^). The effect of pH on stability was evaluated by incubating 300 µL of the enzymatic crude extract in 150 µL of the buffer with pH values ranging from 2 to 8, for 60 min and 12 h, at 8 °C (±3 °C). After the pH treatment, the reaction was performed as described above. 

The optimal temperature of the enzymatic crude extract was assessed in the range between 20 and 80 °C, with 10 °C intervals, by conducting the reaction with the previously selected optimal pH. The effect of the temperature on stability was studied by incubating 300 µL of the enzymatic crude extract in the same temperature range, for 60 min, in water-baths. After the thermic treatment, the reaction was performed as described above.

### 2.3. Proteomic Analysis

To determine the enzymatic composition of the crude extract, a proteomic profiling analysis of soluble proteins was carried out using a shotgun strategy via nano LC-MS/MS analysis followed by a comparison to specific databases according to Santos et al. [25].

### 2.4. Protein Digestion

The proteins were treated with 100 mM dithiothreitol (Sigma-Aldrich, SP, Brazil) at 60 °C for 30 min for protein reduction, then alkylated with 300 mM iodoacetamide (Sigma-Aldrich, SP, Brazil) for 30 min, and digested with trypsin (Promega, Fitchburg, WI, USA) overnight at 37 °C.

### 2.5. Mass Spectrometry: Nano LC-MS/MS

The mixture of peptides was re-suspended in 0.1% formic acid and analyzed in a Q Exactive Plus mass spectrometer (Thermo Scientific, Waltham, MA USA) connected to an Easy nLC 1000 nano-liquid chromatography system (Thermo Scientific, Waltham, MA, USA). Peptides were loaded in a pre-column (2 cm length, 200 µm inner diameter, packed in-house with ReproSil-Pur C18-AQ 5 µm resin—Dr. Maisch GmbH HPLC) and fractionated in a column (25 cm length, 75 µm inner diameter, packed in-house with ReproSil-Gold C18 3 µm resin—Dr. Maisch GmbH HPLC) at a constant flow rate of 250 nL min^−1^. 

A gradient was established using solvents A (95% H_2_O, 5% ACN, 0.1% formic acid) and B (95% ACN, 5% H_2_O, 0.1%formic acid) to elute the peptides. The gradient started with 5% solvent B and increased to 45% during 24 min. In the next 3 min, solvent B increased to 95% and it was kept at 95% for 8 min. 

Samples were analyzed using a top 15 data-dependent acquisition (DDA) method. The operating parameters of the mass spectrometer were: spray voltage at 2.5 kV, zero flow of sheath and auxiliary gas and 250 °C in the heated capillary. MS1 scan was acquired in the Orbitrap analyzer with a *m*/*z* 350–2000 interval, 70,000 (at *m*/*z* 200) resolution, target AGC value of 1 × 10^6^ and maximum injection time of 100 ms. The 15 most intense ions (charge ≥ 2) were selected and fragmented in a higher energy collisional dissociation cell with a normalized collision energy of 30. MS2 spectra were acquired with a 17,500 (at *m*/*z* 200) resolution, target AGC value of 1 × 10^5^ and maximum injection time of 100 ms. The ion selection threshold was set to 3 × 10^5^, and dynamic exclusion was set to 45 s.

### 2.6. Data Analysis

Raw files were processed by the Proteome Discoverer (PD) 2.1 software (Thermo Scientific, Waltham, MA, USA) and spectral data were searched using Sequest HT^®^ algorithm. The UniProt database limited to *S. cerevisiae* (strain ATCC 204508/S288c) reference proteome set was downloaded from UniProt consortium in January 2019 (6729 entries). For a more specific pectinase search, a pectinase database was assembled and downloaded, also from UniProt consortium, comprising of all the pectin active related proteins from Basidiomycota and Ascomycota Subkingdom (16,702 entries) spectral data were searched using Sequest HT^®^ algorithm. The parameters used in PD Sequest HT^®^ were: full-tryptic search space, up to two missed cleavages allowed for trypsin, precursor mass tolerance of 10 ppm, and fragment mass tolerance of 0.05 Da. Carbamidomethylation of cysteine was included as fixed modification, and methionine oxidation and protein N-terminal acetylation were included as dynamic modifications. To estimate the False Discovery Rate (FDR) of <1% and protein grouping, we used the nodes Percolator and Target Decoy PSM and maximum parsimony, respectively. A cutoff score was established to accept a FDR of 1% at the peptide level.

The *S. cerevisiae* identified proteins were annotated according to gene ontology (GO) terms, classified with Blast2GO software (v 3.0) [26] into biological processes, molecular functions, and cellular components under the default parameters [27].

## 3. Results

### 3.1. Fractional Factorial Experimental Design

The Pareto’s chart in Figure 1 enables the verification of the significant variables for the model and to evaluate their trend. The positive sign (+) of the independent variable factor indicates that an increase in its value leads to an increase in the response, while the negative sign (−) indicates that an increase in its value leads to a reduction of the response [28].

PFRF content (variable #1) was by far the most important variable for pectinase secretion in this experiment, with a *p*-value of 17.48. The pH value of the media (#2) was not significant indicating that, within the tested range, it did not alter the response. Results confirmed that the temperature (#3) was a significant variable, indicating increases in temperature resulted in lower enzymatic activities, in the tested interval (21–35 °C). Agitation (#4) was also significant and indicated that lower speeds were more efficient. Among the four significant variables, the supplementation with peptone (#5) presented the lowest value, which was unexpected and lead to an increase in the peptone concentration in further experiments, in order to verify its influence on pectinase activity. 

### 3.2. Central Composite Rotational Design 1 (CCRD-1)

The highest value of total pectinolytic activity achieved among the 17 assays performed was 6.1 U mL^−1^, applying 28 °C, 28 g 100 g^−1^ of PFRF and 1 g 100 g^−1^ of peptone, followed by 5.6 U mL^−1^ at 24.7 °C, 41 g 100 g^−1^ of PFRF and 1.6 g 100 g^−1^ of peptone; a significant increase compared to the highest result obtained in the previous experiment, of 2.3 U mL^−1^. All results may be observed in Appendix A. In Figure 2A,B the influence of the independent variables and the interaction among them on the enzymatic activity may be observed.

The secretion of pectinases by *S. cerevisiae* was affected linearly (but not quadratic) by the PFRF of the medium and in a quadratic manner by the culture temperature (Figure 2B). These results suggest that a higher content of flour would lead to increased pectinolytic activity. A third experimental design (CCRD-2) aiming to optimize the most influential variables and to verify a possible increase in the secretion of the total pectinolytic activity was elaborated, where the variable “peptone content” was eliminated and higher temperature range and flour content were tested.

### 3.3. Central Composite Rotational Design 2 (CCRD-2)

The variance analysis (ANOVA) of the response surface model determined the calculated F value (11.54), a value 2.5 times higher than the tabulated F, which conferred to the model a high significance. The quality of the regression model adjustment was tested evaluating the determination coefficient based on r^2^. The CCRD-2 model presented an elevated regression coefficient that explained 87% of the response variability. The adjusted determination coefficient (Adj r^2^ = 0.74) was also high, showing a good adjustment between the observed and expected responses (Appendix A). The higher value of total pectinolytic activity achieved was 6.1, followed by 5.3 U mL^−1^, with temperatures of 24 and 32 °C, PFRF content of 60 and 52 g 100 g^−1^, respectively. There was no significant increase in the total pectinolytic activity when compared to the previous experiment; however, similar activity values were obtained with the flour ratio increase in the absence of peptone, confirming that supplementation was indeed unnecessary. A second-order polynomial equation (Equation (2)) was applied to correlate the independent variables “PFRF (f, in g 100 g^−1^)” and “culture temperature (t, in °C)” with the dependent variable “total pectinolytic activity (U mL^−1^)”. The resulting surfaces may be observed in Figure 2C.
(2)$$\mathrm{Total}\;\mathrm{pectinolytic}\;\mathrm{activity}\;\left(U\times {\mathrm{mL}}^{-1}\right)=-7.054-0.04f-0.0006f^{2}+0.77t-0.02t^{2}+0.007\mathrm{ft}$$

### 3.4. Effect of pH and Temperature on the Enzymatic Crude Extract Activity and Stability

The effect of the reaction medium’s pH and temperature on the total pectinolytic activity of the crude enzymatic extract may be observed in Figure 3. The optimal pH (Figure 3A) for pectinolytic activity was observed at pH 3; however, there were no significant differences in the pH values between 2 to 5. The enzymatic extract produced in the optimized medium was kept in several pH values for 1 or 12 h (Figure 3C,D, respectively). In both cases, the enzymatic extract kept its residual activity above 70% of the total original pectinolytic activity in the wide pH range tested, indicating excellent stability.

The optimal temperature of the enzymatic extract was obtained at 70 °C (Figure 3B); however, there were no significant differences in the ranges of 30–40 °C and 60–80 °C. This study showed that the pectinolytic crude extract kept more than 80% of its original activity in the whole range tested (except at 60 °C), after 60 min of thermal treatment (Figure 3E). 

### 3.5. Pectinolytic Enzymes Activity

The presence of different pectinolytic enzymes in the crude enzymatic extract secreted in the optimal conditions of fermentation was tested. The PG activity determined in this study was 3.5 ± 0.5 U mL^−1^; the PME activity was of 0.08 ± 3 × 10^−3^ U mL^−1^; the PL activity was of 3.1 ± 0.4 U mL^−1^ and the PTL activity was of 0.8 ± 3 × 10^−2^ U mL^−1^.

### 3.6. Proteomic Characterization

The GO analysis of the 48 identified proteins (the list of identified proteins is presented in the Appendix A) secreted by the *S. cerevisiae* strain in this study is presented in Figure 4. The proteins were classified into cellular component, molecular function and biological process using the Blast2Go software. A total of 492 annotations is depicted, indicating that the same protein may have been assorted into many different classes. Most hits were found for biological process (69%) followed by cellular component. Regarding molecular function, 30% of the annotations were associated to catalytic activity, the class expected to be associated with the pectinolytic enzymes studied.

Analysis using the pectinase database resulted in a total of 11 proteins (presented in Appendix A) of which five were classified as ‘uncharacterized’, probably because they could not be matched to any of the sequences in the databank. Of the nine identified proteins, five were not pectin-related and of the four remaining, two were pectin–lyases, one was a pectate–lyase and one was a glucosidase, as detailed in Figure 5. Just one PG sequence is annotated from the *S. cerevisiae* genome and is found in the *S. cerevisiae* proteome, as downloaded from Uniprot. None of the identified proteins found in the present study showed homology to this described sequence. 

## 4. Discussion

### 4.1. Sequential Strategy of Experimental Designs for the Production of Pectinases by SSF

Industry acceptance and application of enzymatic processes depends upon the cost on a large-scale enzyme production. This work investigated the possibility of obtaining pectinolytic enzymes from a selected *S*. *cerevisiae* strain using an agro-industrial residue substrate as a possible strategy to reduce process cost. 

### 4.2. Fractional Factorial Experimental Design

The variables that affect the enzymatic synthesis and secretion must be observed for each fermentation process, as the optimal conditions vary among the microorganisms’ strains and the target enzymes [29].

The PFRF content, in addition to providing nutrients to the cells, was also decisive in the resulting type of fermentation. Assays with 1 and 6% PFRF consisted of submerged fermentation, while assays with 11% PFRF functioned as SSF. The latter proved to be more efficient for the pectinolytic activity secretion by the *S. cerevisiae* strain in this study.

The pH value of the growth medium is generally an important variable to be considered when cultivating microorganisms, as each strain has an optimal pH for better growth. However, microorganisms have an efficient intracellular buffering capacity, thus the pH of the medium affects mostly the secreted enzymes [30]. According to Biz et al. [31] and Poondla et al. [27] the production of pectinases by yeasts may be maximized when the pH is kept between 4 to 6, therefore this variable was fixed at pH 5 for the subsequent experimental designs. This decision was also supported by Martos et al. [32] and Jayani et al. [33], which found optimal PG activity and stability at pH 4 and 5 for the enzymes secreted by *Wickerhamomy cesanomalus* and *S. cerevisiae*, respectively.

An increase in the cultivation temperature may accelerate chemical and enzymatic reactions in the cells and speed up the cellular multiplication. However, this increase may not exceed the maximum temperature tolerated by the microorganisms. Zakhartsev et al. [34] determined the *S. cerevisiae* optimal temperature range of fermentation as around 31 °C, with an increase in growth from 26 to 31 °C and decrease from 33 to 40 °C. Poondla et al. [23] showed maximum pectinases activity in fermentation at 30 °C and increased activity with temperatures increasing between 4 and 30 °C but showed a reduction of activity at temperatures higher than 35 to 45 °C. Preliminary results in this test seem to indicate a greater retention of enzymatic activity at lower temperatures, which may be related to the low thermal resistance of the enzymes secreted by the strain under study. Thus, the following experimental design tested a temperature range including lower temperatures, up to 12 °C.

The low value for the peptone concentration significance was, to some extent, unexpected as the reports found in the literature strongly suggested nitrogen supplementation as an important variable for large scale enzyme production: Arévalo-Villena et al. [35]; Kaur et al. [36] and Maidana et al. [6] concluded that the addition of yeast extract, peptone, ammonia or urea to agro-industrial residue media significantly increased the levels of secreted enzymatic activity by yeasts. To better evaluate the effect of peptone supplementation of the culture medium, for the subsequent experimental design, the maximum peptone content tested was doubled.

Agitation is considered an important resort in fermentation, as it leads to the homogeneity of the culture medium and incorporation of air [37]. In this study Consequently, static incubation was adopted for the further assays.

### 4.3. Central Composite Rotational Design 1 (CCRD-1)

The peptone content, within the tested range, presented no alteration in the response regardless of the concentration (Figure 2A). Possibly, the use of a higher PFRF content ensured the required amounts of nitrogen for the enzyme secretion and cellular development, making the supplementation with peptone unnecessary. Considering the expected cost difference between peptone and PFRF, a residue from the juice industry, it was preferable to eliminate the use of the former and increase the content of the latter.

### 4.4. Central Composite Rotational Design 2 (CCRD-2)

The enzymatic activity reached in this study could be obtained in a shorter incubation time than reported by studies available with other strains and other sources of agro-industrial residue. The enzymatic activity obtained in a medium containing orange peel and peanut oil in 48 h was 6.3 U mL^−1^ [28]. Further, Poondla et al. [23] achieved the activity of 5.9 U mL^−1^ after 48 h using exclusively beef broth as a nutrient source. Jaramillo et al. [38] determined the activity of a pectinase produced by *Aspergillus niger* to be 0.2 U mL^−1^ when fermented at 28 °C, using 1% of yellow passion fruit peel as carbon source. When compared to an assay with similar flour content (4%), the *S. cerevisiae* strain used in the present study produced both 0.9 U mL^−1^ in CCRD-1 (assay 9) and 2 (assay 5).

There were no significant differences in the total pectinolytic activity when PFRF was applied in proportions ranging from 50 to 60 g 100 g^−1^, with incubation temperatures between 22 and 36 °C, a wide range in which the enzymatic activity was maintained (Figure 2C). Thus, in order to obtain a higher total pectinolytic activity for the lowest production cost, the application of this *S. cerevisiae* strain may be recommended in a culture medium with 50 g 100 g^−1^ of PFRF with water (instead of buffer-controlled pH) and no supplementation. The fermentation may be carried out at room temperature, as long as in the range of 22–36 °C. Independent experiments were performed in triplicate, applying the optimized conditions, in order to validate the model adequacy. The maximum enzyme activity observed was 7.1 ± 0.6 U mL^−1^. The activity observed with the optimized medium showed a 300% increase compared to the higher activity observed in the first experiment (2.3 U mL^−1^).

In the pursuit of low-cost formulations, PFRF was determined as a good resource to be used for pectinase production, acting as a source of carbon and energy to support microbial growth [38], as well as a source of pectin. According to Mohandas et al. [39] and Maidana et al. [6], pectinase activity tends to be significantly higher when the microorganism grows in pectin added medium, in comparison to simple sugars. The passion fruit peel shows a pectin content of around 400 mg g^−1^ of dry matter [15], a high content in comparison to other food residues as reported by Müller-Maatsch et al. [40]. Besides the pectin content, the passion fruit peel flour can reach amounts of ash, proteins, fat, and other carbohydrates of 7.5, 4.8, 0.9, and 23.4% *w*/*w* (dry basis) respectively, according to Duarte et al. [13]. In light of the results obtained for both variables “PFRF” and “agitation”, and considering the advantages involved in the application of more concentrated (solid) culture media, the subsequent designs were tested to a flour content up to 50%, resulting in solid culture media.

### 4.5. Effect of pH and Temperature on the Enzymatic Crude Extract Activity and Stability

The medium pH is an important reaction variable, as it can lead to conformational modifications of the enzyme active site, usually causing a change in activity by altering the affinity to the substrate [41,42]. Overall, the activity showed small variations in the tested interval, which suggests that the enzymatic extract may be composed of different isoenzymes, that may present distinct optimal pH values, ensuring a good total pectinolytic activity in a wide pH range. Pectinases produced by fungi generally show optimal pH in the acid range (pH 4–5), slightly varying according to the microorganism [23,43]. Considering that most fruits in the juice industry show pH values lower than 4, the production of pectinolytic enzymes with optimum activity at pH 3 may offer an interesting alternative for application in this industrial sector. The pH affects the stability of the enzymes because the lateral chains of several amino acids act as weak acids or bases. Therefore, when subjected to extreme pH values, both acid and alkaline, enzymes may undergo denaturation, even irreversibly. The crude extract in this study kept its activity above 70% between pH values from 2 to 8. This behavior may also result from the presence of multiple isoenzymes, each of them with optimal stability in one pH range and contributing to the maintenance of the overall pectinolytic activity. Similarly, Bennamoun et al. [43] reported 90% stability for an exo-polygalacturonase, secreted by a *Aureobasidium pullulans* strain, in a wide pH range (4–10) for up to 3 h.

The results of the influence of the temperature on the total pectinolytic activity (Figure 3A) also suggests the presence of isoforms, with optimal temperatures at 30–40 °C and 70 °C. The literature presents optimal temperatures for yeast pectinases in the range of 30–60 °C, depending on the producing microorganism [23,43]. The present study showed optimal activity in higher temperatures when compared to other studies, a valuable characteristic, as biotechnological processes performed in high temperatures offer a significantly lower risk of microbial contamination. The increase in temperature favors the reaction speed, since it offers more kinetic energy and, consequently, increases the number of collisions between molecules (enzyme and substrate) [44,45]. However, typically, the temperature increase above specified values for each enzyme, increases their molecular agitation, breaking the intermolecular bonds responsible for the proteins’ superior structures causing denaturation and an abrupt interruption of the activity. Evidence of pectin active isoforms produced by yeast strains have been provided by Barnby et al. [46] studying a strain of *Kluyveromyces marxianus*. 

The literature presents examples of pectin active enzymes with high optimal temperatures and also high stability. The optimum temperature of an exo-polygalacturonase secreted by *A. pullulans* was found to be 60 °C, which similarly presented more than 70% stability within the range of 60–90 °C for 1 h [43]. In comparison to a pectinase secreted by *Geotrichum candidum*, which presented around 60% residual activity in the range of 30–40 °C, the pectinase in this study showed higher thermal resistance and the possibility to be employed in high temperature processes (up to 80 °C) for at least one hour.

### 4.6. Pectinolytic Enzymes Activity

Among the different pectinases, PG is the most common enzyme industrially applied and mainly produced by filamentous fungi, which demand higher fermentation periods in comparison to yeasts like *S. cerevisiae.* [47]. SSF is usually employed for filamentous fungi the PG activity of which is generally several times higher than that from yeasts. The fermentation of orange peel by an *Aspergillus sujae* strain, for 8 days, lead to 145 U mL^−1^ PG activity, and the fermentation of apple pomace by a *Penicillium expansum* strain, for four days, achieved 1103 U mL^−1^ PG activity [26,48]. Even yeasts, after longer fermentations, were able to generate higher PG activities when compared to the present study. Orange peel fermentation by a *Zygoascus hellenicus* strain, for 48 h, generated 29 U mL^−1^ PG activity whereas tomato pomace fermentation by an *Aureobasidium pullulans* strain, for 72 h, lead to 26 U mL^−1^ PG activity [43,49]. Together with the higher PG activity by filamentous fungi, a higher fermentation time is required, and despite the lower activity from yeasts, the decreased process time might prove overall appealing.

According to Poondla et al. [23], the PME production by yeasts is normally reduced when compared to PG. Low PME activity is interesting for minimizing the release of methanol in the reaction medium, and even in the absence of PME, the highly methoxylated pectin may be lyzed by PL [1]. 

After an extensive literature search regarding pectinase secretion by *S. cerevisiae*, two studies alone pointed out the presence of PL and none made any reference to the secretion of PTL. Gainvors et al. [50] (1994) identified PG and PME activity besides PL and the findings by Poondla et al. [23] (2015) also included PME and PL activity, this latter in values similar to those found in this study (4.1 U mL^−1^), but lower than the activity (46 U mL^−1^) secreted by the filamentous fungi strain of *Aspergillus brasiliensis*, after 124 h fermentation of orange peel [51] (Pili et al., 2018).

The production of PTL by yeast strains is rarely mentioned in the literature. In fact, extensive research found only a reference to a strain of *Debaryomyces nepalensis* a halotolerant food-spoiling yeast species [52].

### 4.7. Proteomic Characterization

The present study applied the most common way of identifying proteins from peptide mass spectra: the database search method. In this method, proteins with known amino acid sequences are digested ‘in silico’ and the peptides generated are transformed into hypothetical mass spectra. When a mass spectrum of a real peptide is similar to a spectrum belonging to a database, the peptide is identified. The search for similar mass spectra is done by algorithms, which use scoring methods to guarantee the reliability of the identification. A successful identification, therefore, depends on the use of databases that contain the protein sequences (and their peptides) of the proteins that are in the sample [20]. Little is known about the existence of PTLs produced by yeasts, as well as few databases containing sequences of these enzymes are available, which makes it extremely difficult to identify the pectinases present in the culture medium produced in this study. This does not mean that enzymes cannot be produced. Paulo et al. [29] demonstrated that different carbon sources may generate different responses in *S. cerevisiae* metabolism. In their study, the proteins related to metabolic function and transmembrane transport underwent the most considerable changes in response to carbon source variation. Thus, the pectin-rich medium optimized in this study may have induced a restricted enzyme production, focusing the secretion on pectin active enzymes.

Another significant finding was that despite of PG being the highest pectinolytic activity experimentally found, the three identified enzymes matched to the pectinase database were lyases, one of them a PTL. These results might be related to the low efficiency of the methods for lyase activity detection applied, that relied on UV absorption at 235 nm of the double bonds of the reaction products. Even though the structures of the PTL were not elucidated in this study, LC-MS/MS results corroborate the experimentally tested pectinase activity, until now not reported for *S. cerevisiae*.

## Figures and Tables

**Figure 1 molecules-27-04981-f001:**
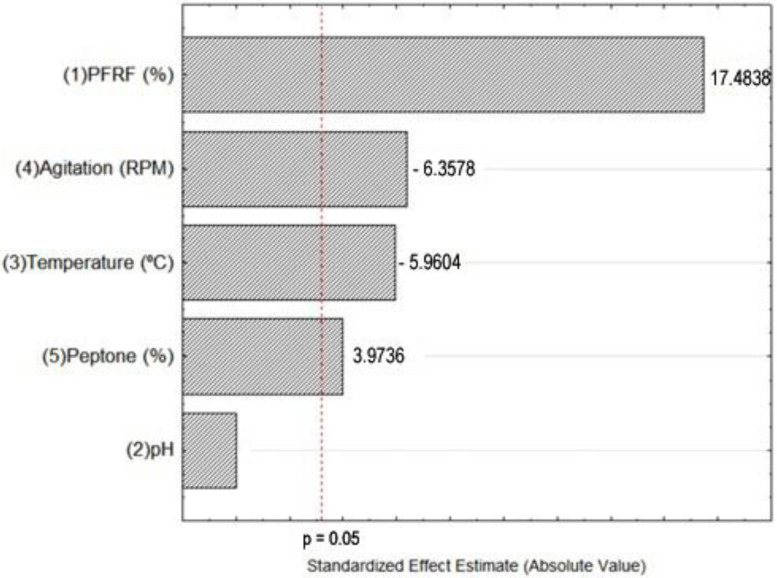
Pareto’s chart for the fractional factorial experimental design.

**Figure 2 molecules-27-04981-f002:**
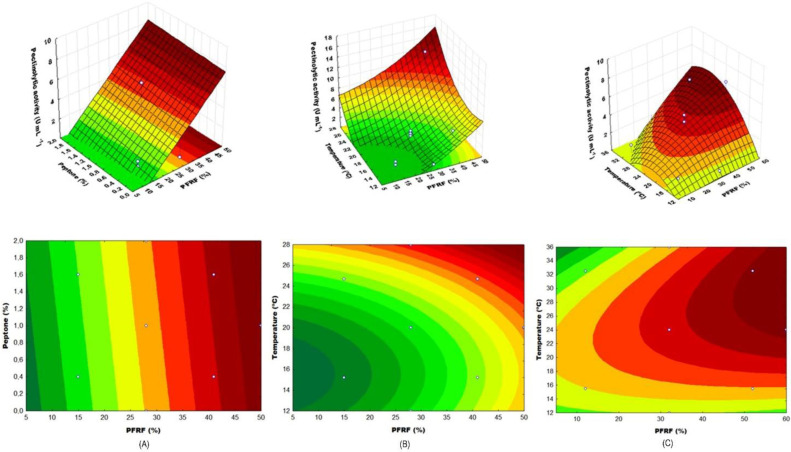
Response surfaces and contour curves for the influence of the tested variables in the secretion of total pectinolytic activity (U mL^−1^) by *S. cerevisiae*. (**A**) peptone content and PFRF content (g 100 g^−1^)—CCRD-1; (**B**) temperature (°C) and PFRF content (g 100 g^−1^)—CCRD-1; (**C**) temperature (°C) and PFRF content (g 100 g^−1^)—CCRD-2.

**Figure 3 molecules-27-04981-f003:**
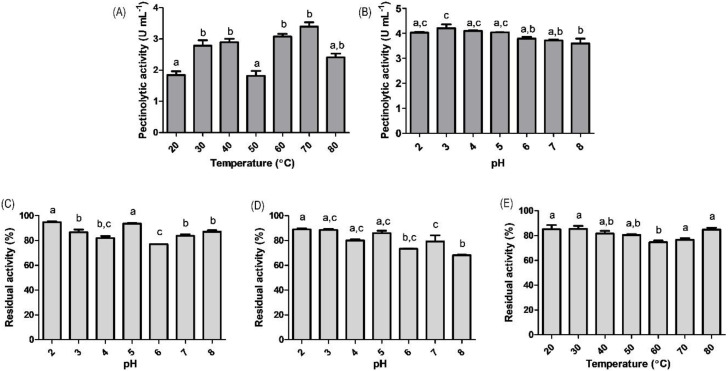
Influence of (**A**) pH and (**B**) temperature in the activity and in the stability after (**C**) 1 h and (**D**) 12 h pH treatment and (**E**) 1 h temperature treatment of the crude pectinolytic extract. Lowercase letters indicate significant differences between treatments (*p* < 0.05).

**Figure 4 molecules-27-04981-f004:**
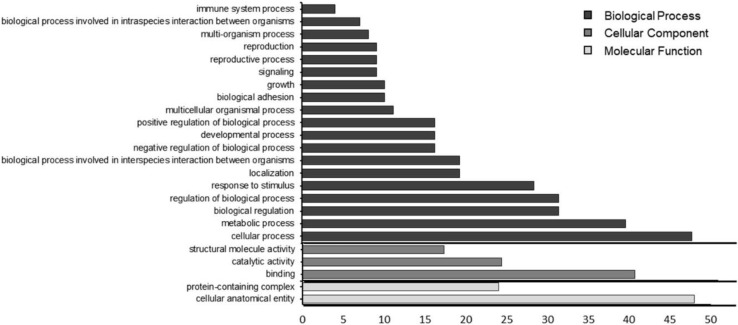
Functional classification of proteins from the pectinase crude extract secreted by *S. cerevisiae* strain.

**Figure 5 molecules-27-04981-f005:**
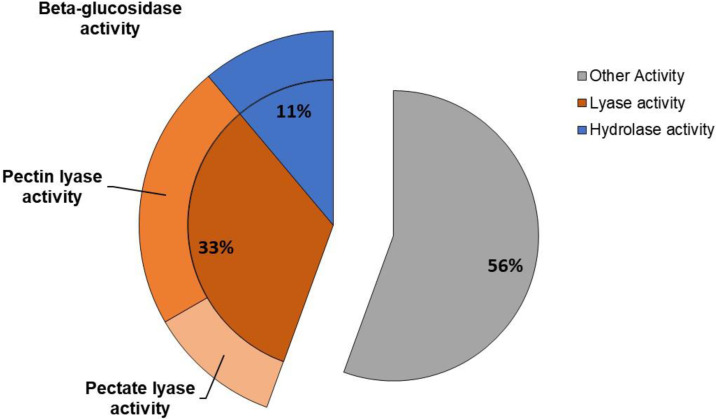
Percent distribution of the number of pectin active enzymes in the pectinase crude extract secreted by *S. cerevisiae* strain.

**Table 1 molecules-27-04981-t001:** Independent variables and coded and noncoded values used in the sequential strategy of experimental designs aiming an increased secretion of pectinase.

	Coded Variable (Level)	pH	Agitation (rpm)	Peptone (% *w*/*w*)	Temperature (°C)	PFRF (% *w*/*w*)
Fractional factorial	−1	5	0	0.0	21.0	1.0
	0	7	100	0.5	28.0	6.0
	+1	9	200	1.0	35.0	11.0
CCRD 1	−α (−1.68)	-	-	0.0	12.0	4.0
	−1	-	-	0.4	15.2	15.0
	0	-	-	1.0	20.0	28.0
	+1	-	-	1.6	24.7	41.0
	+α (+1.68)	-	-	2.0	28.0	50.0
CCRD 2	−α (−1.41)	-	-	-	12.0	4.0
	−1	-	-	-	15.5	12.0
	0	-	-	-	24.0	32.0
	+1	-	-	-	32.5	52.0
	+α (+1.41)	-	-	-	52.0	60.0

## Data Availability

The data presented in this study are available on request from the corresponding author.

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
