# Peer review of "Pectinases Secretion by Saccharomyces cerevisiae: Optimization in Solid-State Fermentation and Identification by a Shotgun Proteomics Approach"

_molecules, 2022, doi:10.3390/molecules27154981_

Round 1

Reviewer 1 Report

The current manuscript submitted to MOLECULES MDPI,  investigated the hypothesis of obtaining pectinolytic enzymes from a selected S. cerevisiae strain using an agro-industrial residue substrate as a possible strategy to reduce the processing costs. The use of certain enzymes in different foodstuffs is of great importance for the food industry, as these can provide the products with beneficial health and special sensory properties.

In addition, the article is novel and original in nature given that the authors documented the synthesis of pectate lyase by S. cerevisiae.

The optimal physicochemical parameters were highlighted by chemometrics. The English language of the article is very good and the paper has a very well-built flow.

I have indicated within the attached pdf some corrections for the authors. Based on these comments, I suggest a minor revision prior to further consideration for publication.

Author Response

On behalf of all authors, I would like to thank the reviewer for the careful reading of our manuscript and for all the comments.

Response. We have deleted and corrected the highlighted words in the manuscript. 

Regards,

MGBKoblitz

Reviewer 2 Report

Dear Authors, I would like to appreciate the study conducted to produce pectinases by S. cerevisiae on passion fruit residue flour through solid-state fermentation. I suggest minor revision of manuscript before publication in Molecules.

1. Line 74, mention if your study is the first in this field, if not, mention novelty of your study by comparing with previous studies reported in the field.

2. Line 121, what is item 2.4.

3. Line 141, in equation 1, what A.M stands for.

4. Line 159-160, 198, and 202, Check the format wherever applicable in the manuscript.

Author Response

On behalf of all authors, I would like to thank the reviewer for the careful reading of our manuscript and for all the comments. We provide below a point-by-point response.

1. Line 74, mention if your study is the first in this field, if not, mention novelty of your study by comparing with previous studies reported in the field.

Response 1. We have included the phrase "for the first time to the extent of our knowledge” to line #78.

2. Line 121, what is item 2.4.

Response 2. We changed "item 2.4" to "in 'Enzyme Assays' below"

3. Line 141, in equation 1, what A.M stands for.

Response 3. AM should mean “mono galacturonic acid” but this information was unnecessary to the equation and was deleted.

4. Line 159-160, 198, and 202, Check the format wherever applicable in the manuscript.

Response 4. We checked the format throughout the manuscript.

Regards,

MGBKoblitz.